# The Influence of Dysphagia on Nutritional and Frailty Status among Community-Dwelling Older Adults

**DOI:** 10.3390/nu13020512

**Published:** 2021-02-04

**Authors:** Takahiro Nishida, Kazumi Yamabe, Sumihisa Honda

**Affiliations:** 1Sasebo-Yoshii Community Comprehensive Support Center, Sasebo 859-6305, Japan; taka444c@yahoo.co.jp; 2Department of Public Health Nursing, Nagasaki University Graduate School of Biomedical Sciences, Nagasaki 852-8520, Japan; 3Yamabe Dental Clinic, Sasebo 859-6101, Japan; taku201@mocha.ocn.ne.jp

**Keywords:** nutritional status, frailty, dysphagia, community-dwelling older adults

## Abstract

Malnutrition is a core symptom of the frailty cycle in older adults. The purpose of this study was to investigate whether dysphagia influences nutrition or frailty status in community-dwelling older adults. The study participants were 320 Japanese community-dwelling older adults aged ≥65 years. All participants completed a questionnaire survey that included items on age, sex, family structure, self-rated health, nutritional and frailty status, and swallowing function. Nutritional status was categorized as malnourished, at risk of malnutrition, and well-nourished based on the Mini Nutrition Assessment-Short Form. The participants were then classified into a malnutrition (malnourished/at risk) or a well-nourished group (well-nourished). Frailty was assessed using the Cardiovascular Health Study criteria. The participants were then divided into a frailty (frail/pre-frail) or a non-frailty group (robust). Dysphagia was screened using the 10-item Eating Assessment Tool. Multiple logistic regression analysis was conducted to determine whether dysphagia was associated with nutritional or frailty status. The results revealed that dysphagia influenced both nutrition (odds ratio [OR]: 4.0; 95% confidence interval [CI]: 1.9–8.2) and frailty status (OR: 2.3; 95% CI: 1.0–5.2); therefore, the swallowing function would be an important factor for community-dwelling older adults on frailty prevention programs.

## 1. Introduction

Frailty prevention is being increasingly implemented in aging societies such as Japan, where the prevalence of frailty among the older population is reported to be around 10% [1,2]. Frailty has been defined as a vulnerable state to stressors resulting from age-related declines in function and reserve across multiple physiological systems that compromise the ability to maintain homeostasis [3]. The components of the clinical phenotype of frailty consist of the following five domains: low strength, slow motor performance, exhaustion, low physical activity, and unintentional weight loss [4]. The core of the frailty cycle is considered to be malnutrition, which can cause a vicious cycle of appetite loss owing to muscle loss, muscle weakness, fatigue, and decreased energy expenditure [4]. Since frailty is expected to be modified by early detection and intervention, frailty prevention programs at a public hall are recommended for community-dwelling older adults in Japan [1,2]. As a clinical condition, malnutrition has been defined as an imbalance of energy, protein, and other nutrients that causes negative effects on body composition, physical function, and clinical outcomes [5]. The prevalence of undernutrition status is estimated to be as high as 25–35% among community-dwelling older adults [6,7]. Therefore, sufficient nutritional intake, such as protein and energy, is recommended in combination with training for muscle strengthening, such as resistance training, in frailty prevention programs among older people [8].

Recently, the concept of preventing oral frailty has become popular in Japan [9,10]. Oral frailty is the successor to “8020 campaign” that started since 1989 and focused mainly on the prevention of tooth loss [11]. By contrast, oral frailty focuses on preventive care for oral dysfunction, such as swallowing ability [9,10]. Swallowing ability is a crucial oral function because it is a major risk factor for aspiration pneumonia in older people [12,13]. Moreover, the prevalence of dysphagia among community-dwelling older people has increased by approximately 25% [14,15].

Many previous studies have reported that dysphagia can cause malnutrition and adverse health outcomes in clinical settings [16,17,18]; however, the effects of dysphagia on nutritional status in the community setting remain poorly understood. Furthermore, although malnutrition is associated with physical frailty and considered to the core of the frailty cycle, malnutrition and physical frailty are also thought to have fundamentally distinct pathologies [7]. Frailty is consistent with malnutrition in the pathology of muscle loss and chronic inflammation, but frailty is thought to have more multidimensional effects in older people [7]. Therefore, the objective of this study was to investigate whether dysphagia influences nutrition or frailty status in community-dwelling older adults.

## 2. Materials and Methods

### 2.1. Procedures and Participants

We carried out a cross-sectional study in the local area of Sasebo city, Nagasaki Prefecture, Japan, between April 2018 and November 2019. As of October 2018, the total population of the study area was 18,651, and that of persons ≥ 65 years was 6787 (aging rate: 36.4%). The participants were consecutively recruited from among community-dwelling older adults aged ≥65 years who were participating in voluntary circle activities for preventive care and health promotion at public halls. All participants could walk, eat and drinking orally, and speak without assistance. Therefore, the activities of daily living (ADLs) of almost all participants were considered to be relatively high. Those aged <65 years were excluded from the analysis.

This study was approved by the Ethics Committee of Nagasaki University Graduate School of Biomedical Sciences (Approval number:19050901). Informed consent was obtained from all participants.

### 2.2. Measurements

We collected data on all potential participants’ basic characteristics (age, sex, and family structure), self-rated health, nutritional status, physical frailty, and swallowing function. Age was divided into 65–74 years (young-old) and ≥75 years (old-old). Family structure was divided into whether participants were living alone or cohabitating. Self-rated health was assessed using the question “How would you rate your current overall health”. The participants gave responses to each question item on a Likert scale (i.e., very good, good, intermediate, poor, and very poor), and were categorized as either “healthy” (responses of very good and good) or “poor health” (responses of intermediate, poor, and very poor). Anthropometric data such as hand grip strength and walking speed were measured by a public health nurse and a physical therapist. The public health nurse had received adequate training on physical fitness tests from the physical therapist before the study began.

### 2.3. Mini Nutritional Assessment-Short Form (MNA-SF)

Nutritional status was evaluated using the Mini Nutritional Assessment-Short Form (MNA-SF) [19], which is composed of six questions related to (1) oral intake within the past 3 months, (2) weight loss within the past 3 months, (3) mobility, (4) stress and/or acute disease within the past 3 months, (5) neuropsychological problems, and (6) body mass index (BMI). The total score on the MNA-SF ranges from 0 to 14 points, and was divided into the following three categories: malnutrition (0–7 points), at risk of malnutrition (8–11 points), and well-nourished (12–14 points). In addition, nutritional status was dichotomized into a malnutrition group (at risk of malnutrition and malnutrition: 0–11 points) and a well-nourished group (well-nourished: 12–14 points).

### 2.4. Cardiovascular Health Study (CHS)

Frailty was evaluated based on the Cardiovascular Health Study (CHS) criteria originally reported by Fried et al. [3]. The CHS criteria are composed of five domains: slowness, weakness, exhaustion, low activity, and weight loss. In this study, we adopted the Japanese version of the CHS criteria in reference to a previous report [20]. Slowness was measured using a stopwatch and determined by a comfortable 5 m walking speed. The cutoff point for slowness was <1.0 m/s. Weakness was evaluated based on hand grip strength and measured using a Smedley-type handheld dynamometer (Tsutsumi Co., Ltd., Tokyo, Japan). The cutoff point for dominant hand grip strength was <26 kg for men and <18 kg for women. Walking speed and hand grip strength were measured twice, with the better values taken as the representative ones. “Exhaustion” and “unintentional weight loss” were considered to be present if the participants answered “Yes” to the following questions, respectively: “In the past 2 weeks, have you felt tired for no reason?” and “Have you lost more than 4–5 kg unintentionally in the past 1 year?” “Low activity” was considered to be present if the participants answered “No” to the following question: “Do you engage in physical exercise or sports aimed at improving health at least once a week?” One point was assigned for the presence of each of the five domains, and then the participants were categorized according to total points as follows based on the CHS phenotype model: frail (3–5 points), pre-frail (1–2 points), and robust (0 points). In this study, we combined frail and pre-frail. Therefore, the phenotype model was dichotomized into the frailty group (pre-frail and frail: 1–5 points) and the non-frailty group (robust: 0 points).

### 2.5. The 10-Item Eating Assessment Tool (EAT−10)

Swallowing function was assessed using the 10-Item Eating Assessment Tool (EAT−10), the reliability and validity of which were demonstrated by Belafsky et al. [21] and Wakabayashi et al. [22]. The EAT−10 is composed of 10 questions on swallowing function rated on a 5-point Likert scale (from 0 = no problem to 4 = severe problem). The total score on the EAT−10 ranges from 0 to 40 points, with a higher score indicate severe dysphagia. The cutoff score for dysphagia on the EAT−10 is 3 points. We classified the participants as with dysphagia or without dysphagia in accordance with previous studies [21,22].

### 2.6. Statistical Analysis

The participants’ basic attributes, including age, sex, and family structure, were reported as number with percentage or mean with standard deviation. The chi-squared test was used to demonstrate the associations between nutritional or frailty status and dysphagia and to compare the well-nourished and malnutrition groups and non-frailty and frailty groups in regard to basic attributes (age, sex, and family structure), self-rated health, and dysphagia. Furthermore, multiple logistic regression analysis was performed to examine whether dysphagia was independently associated with malnutrition and frailty, and adjusted odds ratios (ORs) with 95% confidence intervals (CIs) were calculated. The covariates selected in the model were age, sex, family structure, and self-rated health. All statistical analyses were performed using SPSS software (version 23.0 for Windows; IBM, Tokyo, Japan).

## 3. Results

### 3.1. Characteristics of the Study Participants

A total of 358 persons who were participating in circle activities were enrolled, among whom 10 were aged <65 years and thereby excluded. Thus, a total of 348 potentially eligible individuals were included in the study, among whom, a further 28 with missing values were excluded. Finally, 320 participants (52 men, 268 women; mean age: 77 years) were analyzed.

The characteristics of the 320 study participants are shown in Table 1. In this study, women and older adults aged ≥75 years (old-old) accounted for about 84% and 65% of the sample, respectively. Regarding nutritional status according to the MNA-SF, the numbers and percentages of those classified as malnutrition, at risk of malnutrition, and well-nourished were nine (2.8%), 84 (26.3%), and 227 (70.9%), respectively; therefore, the malnutrition group consisted of 93 (29.1%) individuals. Similarly, regarding the frailty phenotype model according to the CHS criteria, the numbers and percentages of those classified as frail, pre-frail, and robust were 45 (14.1%), 154 (48.1%), and 121 (37.8%), respectively; therefore, the frailty group consisted of 199 (62.2%) individuals. Regarding swallowing function, 38 (11.9%) participants had dysphagia according to the EAT−10.

### 3.2. Comparison of Characteristics for Nutritional and Frailty Status

Comparisons of the two groups in terms of nutritional status as evaluated by the MNA-SF, frailty status as evaluated by the CHS criteria, and all characteristics, including dysphagia, as evaluated by the EAT−10 are shown in Table 2. The results showed that the malnutrition group had significantly higher proportions of old-old, poor health, and dysphagia compared with the well-nourished group (*p* = 0.020, *p* = 0.039, and *p* < 0.001, respectively). Regarding frailty status, the frailty group had significantly higher proportions of old-old, living alone, poor health, and dysphagia compared with the well-nourished group (*p* < 0.001, *p* = 0.019, *p* < 0.001, and *p* = 0.003, respectively).

### 3.3. Independent Predictors for the Malnutrition and Frailty Groups

The results of the multiple logistic regression analysis for associations with malnutrition and frailty are shown in Table 3. Regarding nutritional status, dysphagia was independently associated with the malnutrition group (OR: 4.0; 95% CI: 1.9–8.2; *p* < 0.001). Regarding frailty status, age (OR: 4.0; 95% CI: 1.5–10.7; *p* = 0.005) and dysphagia (OR: 2.3; 95% CI: 1.0–5.2; *p* = 0.045) were independently associated with the frailty group.

## 4. Discussion

From the perspective of distinguishing pathological conditions between malnutrition and physical frailty, we investigated the associations between swallowing function and nutritional or frailty status among community-dwelling older adults aged ≥ 65 years. In this study, we screened for dysphagia using the EAT−10, malnutrition using the MNA-SF, and frailty using the CHS criteria. The results indicated that dysphagia was significantly associated with malnutrition and frailty.

Older individuals who need support in carrying out ADLs are considered to progress to the disability stage gradually through the frailty stage [23]. In this study, we did not include individuals with a disability stage such as non-ambulatory (i.e., all study participants could walk without assistance or the need for a wheelchair). The frail and pre-frail stages are considered reversible, and receiving appropriate treatment has been reported to slow the progression to the disability stage [1,8,15]. This study enrolled community-dwelling older adults aged ≥65 years, among whom, the rates of frailty, dysphagia, and malnutrition were relatively high, at 14%, 12%, and 29%, respectively; however, these figures are consistent with previous studies [7,9,24]. Moreover, the results of this study showed that among participants with dysphagia, malnutrition accounted for 18.4% and at risk of malnutrition 42.1% (60.5% for the malnourished group), and frail accounted for 28.9% and pre-frail 55.3% (84.2% for the frailty group; data not shown). Importantly, this high prevalence and our finding of its associations indicate that dysphagia may trigger malnutrition and frailty, which has important implications for community strategies to prevent frailty among older populations.

Regarding the associations between nutritional status and swallowing function, in our earlier report, the Kihon Checklist (KCL, a frailty checklist) was used to examine the association between dysphagia and malnutrition in community-dwelling older Japanese adults, but no significant association was found [23]. The KCL’s definitions of dysphagia and malnutrition involve an affirmative answer to the question “Have you choked on tea or soup recently?” and as having a “BMI < 18.5 kg/m^2^ and weight loss of 2–3 kg in the previous 6 months”, respectively [23]. Although the KCL is a beneficial questionnaire that can assess multiple frailty domains in addition to dysphagia and malnutrition, such as motor function, cognitive function, and ADL level, the validity of a specific domain for dysphagia or malnutrition has not been confirmed [25]. By contrast, in this study, we used validated screening tools, the MNA-SF and EAT−10 [19,21,22], to evaluate nutritional status and dysphagia, which suggests that the results of the present study are more reliable than those of our previous study.

The results of our multiple logistic regression analysis showed that dysphagia was independently associated with both frailty and malnutrition, while age was significantly associated with only frailty. A previous study of meta-analysis revealed that older age was a risk factor for frailty among community-dwelling older adults [26]. Our findings that the risk of frailty was associated with age are consistent with the results of the meta-analysis. The findings suggest that age-related differences in frailty and malnutrition would be significant in older adults.

In the case that malnutrition starts the frailty cycle, dysphagia could cause malnutrition and influence weight loss because older adults living in the community receive nutrition (i.e., protein and energy) orally [6,26]. In addition, because malnutrition can cause many adverse effects among older people, such as anemia, falls, predisposition to infection, depression, and sarcopenia [18,24,27], undernutrition prevention programs should incorporate health education strategies. Furthermore, our findings indicate that swallowing function was associated with nutritional status. Preventive swallowing rehabilitation would be useful in health programs for community-dwelling older adults.

The strength of this study lies in the new and important finding that dysphagia was independently associated with malnutrition and frailty status among community-dwelling older adults. However, this study also had some limitations. First, because of the nature of the cross-sectional design, causal relationships could not be determined. Second, our study included a possible selection bias; therefore, the results cannot be fully generalized to other community-dwelling older adults. In fact, only a 16% of the participants were men. Moreover, in this study, the number of “frail” in the CHS phenotype and “malnourished” in terms of nutritional status (MNA-SF) was relatively small, we combined “frail with “pre-frail” and “malnourished” with “at risk of malnutrition”. However, the demographics of the present study participants were relatively similar to those of general community-dwelling older adults [2,7,9,20]. To confirm these results, a follow-up study involving randomly selected participants is needed. Third, this study included some potential confounding factors, such as comorbidities (e.g., stroke, dementia), or medications, that could have affected both malnutrition and frailty [28]. In this study, considering that frailty is a multifactorial syndrome, we adjusted self-rated health for comorbidities. However, the effects of medication and comorbidities need to be investigated in more detail, such as by using the Cumulative Illness Rating Scale for comorbidities. Finally, we did not collect information on the contents of meals, which could affect dysphagia and nutritional status. Therefore, further studies are needed to adjust for these confounding factors.

## 5. Conclusions

In conclusion, the results of this study revealed that dysphagia as screened using the EAT−10 was independently associated with nutritional status as screened by the MNA-SF and frailty status as assessed by CHS criteria among community-dwelling older adults aged ≥65 years. These findings suggest that swallowing function is important factor in frailty prevention programs for community-dwelling older adults. Further intervention trials were needed to identify the causal relationships between dysphagia with frailty and nutritional status among community-dwelling older adults.

## Figures and Tables

**Table 1 nutrients-13-00512-t001:** Characteristics of the study participants (*n* = 320).

Variable	*n*	%	Mean	SD
Sex (woman)	268	83.8		
Age, years			77.3	6.6
Aged ≥ 75 years	210	65.6		
Family structure (living alone)	99	30.9		
Self-rated health (poor health)	192	60.0		
BMI, kg/m^2^			23.2	3.4
Hand grip strength, kg			22.6	7.1
Walking speed, m/s			1.1	0.3
Nutritional status (MNA-SF)				
Well-nourished	227	70.9		
At risk of malnutrition	84	26.3		
Malnourished	9	2.8		
Frailty phenotype model (CHS)				
Robust	121	37.8		
Pre-frail	154	48.1		
Frail	45	14.1		
Dysphagia (EAT−10 score: ≥3)	38	11.9		

SD, standard deviation; BMI, Body Mass Index; MNA-SF, Mini Nutritional Assessment-Short Form; CHS, Cardiovascular Health Study; EAT−10, 10-item Eating Assessment Tool.

**Table 2 nutrients-13-00512-t002:** Comparison of characteristics for nutritional and frailty status.

	Well-Nourished Group (*n* = 227)	Malnutrition Group (*n* = 93)		Non-Frailty Group (*n* = 121)	Frailty Group (*n* = 199)	
Characteristics	*n* (%)	*n* (%)	*P* ^1^	*n* (%)	*n* (%)	*P* ^1^
Sex			0.71			0.095
Man	38 (16.7)	14 (15.1)		25 (20.7)	27 (13.6)	
Woman	189 (83.3)	79 (84.9)		96 (79.3)	172 (86.4)	
Age			0.02			<0.001
Old-old	140 (61.7)	70 (75.3)		61 (50.4)	149 (74.9)	
Young-old	87 (38.3)	23 (24.7)		60 (49.6)	50 (25.1)	
Family structure			0.26			0.019
Living alone	66 (29.1)	33 (35.5)		28 (23.1)	71 (35.7)	
Cohabitating	161 (70.9)	60 (64.5)		93 (76.9)	128 (64.3)	
Self-rated health			0.039			<0.001
Poor health	128 (56.4)	64 (68.8)		55 (45.5)	137 (68.8)	
Healthy	99 (43.6)	29 (31.2)		66 (54.5)	62 (31.2)	
Dysphagia			<0.001			0.003
EAT−10 score: ≥ 3	15 (6.6)	23 (24.7)		6 (5.0)	32 (16.1)	
EAT−10 score: < 3	212 (93.4)	70 (75.3)		115 (95.0)	167 (83.9)	

^1^ Chi-squared test; EAT−10, 10-item Eating Assessment Tool.

**Table 3 nutrients-13-00512-t003:** Independent predictors for the malnutrition and frailty groups among older adults aged ≥65 years.

	MNA-SF (Malnutrition Group)			CHS Criteria (Frailty Group)		
Variable	OR	95% CI	*p*	OR	95% CI	*p*
Sex, man	0.9	0.4–1.8	0.791	1.3	0.5–3.1	0.581
Age, old-old	1.6	0.9–2.7	0.129	4.0	1.5–10.7	0.005
Family structure, living alone	1.3	0.7–2.2	0.399	1.9	0.9–3.7	0.076
Self-rated health, poor health	1.4	0.8–2.4	0.199	1.7	0.8–3.6	0.145
Dysphagia, EAT−10 score: ≥ 3	4.0	1.9–8.2	<0.001	2.3	1.0–5.2	0.045

OR, odds ratio; CI, confidence interval; MNA-SF, Mini Nutritional Assessment-Short Form; CHS, Cardiovascular Health Study; EAT−10, 10-item Eating Assessment Tool.

## Data Availability

The data presented in this study are available on request from the corresponding author. The data are not publicly available due to the need to maintain participant confidentiality.

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
