# Peer review of "The Influence of Dysphagia on Nutritional and Frailty Status among Community-Dwelling Older Adults"

_nutrients, 2021, doi:10.3390/nu13020512_

Round 1

Reviewer 1 Report

Dear Authors, 

thank You so much for giving me a chance to read this Your manuscript. 

I read it with interest, and found that it had a high quality of presentation. 

Your cross-sectional study highlighted that dysphagia was significantly associated with malnutrition and frailty.

Minor comments:

a) the relationship between malnutrition and physical frailty deserves further discussion. Indeed, what You wrote in lines 41-44 and in lines 65-68 was unclear;

b) In Statistical Analysis, one of the covariates selected in the model was self-rated health. Using an objective covariate such as the Cumulative Illness Rating Scale would give your study higher scientific value;

c) Table 3 shows that dysphagia was independently associated with the malnutrition group. Regarding frailty group, age (> 75 y.o.) and dysphagia were independently associated. Age > 75 yo was not independently associated with the malnutrition group. This relevant point should be discussed.        

Reviewer 2 Report

Thank you for asking me to review this paper describing a cross-sectional cohort study of Japanese community dwelling older adults which showed an association between dysphagia (measured with the EAT-10 score) with malnutrition and frailty scores. Generally, the methods and results are sound and presented concisely, but unfortunately the authors read too much into their findings suggesting their results indicate that swallow rehabilitation should be incorporated into health programs. This would need an intervention trial. I have provided some specific comments below

Introduction

Generally well written, no specific comments

Methods

2.1, line 84 to 90 – these are results not methods

Otherwise the methods are very clear

Results

Men seem underrepresented (17%). Deserves comment in the discussion.

Due to the way in which the participants were selected (approaching people attending health promotion groups in public halls), relative few seem to be accrued in the frail group (n=49), and for analyses, the investigators have combined the pre-Frail group with the Frail group, which could be misleading. Deserves some comment

Table 2. The p-values for the nutritional status groups do not align and therefore difficult to work out which characteristics they relate to (i.e. the 0.71 on the final row of table).

Not all of the data is presented, as described in methods: grip strength, walking speed (accepting that these parameters helped determine overall frailty status), BMI.

Many of the parameters have been categorised, whereas it would be helpful to the reader so see the distribution of the data (means/sd; medians/IQR)

No information on co-morbidities that can affect swallow – e.g. stroke, dementia

Discussion

The following statements is cannot be justified from a cross sectional study:

 “Furthermore, our findings indicate that swallowing function affects nutritional status, which suggests that preventive swallowing rehabilitation should be also incorporated into health programs for community-dwelling older adults”….[This needs an intervention trial]

And

…” These findings suggest that a decline in swallowing function may promote the deterioration of nutritional status and trigger the frailty cycle.”  [as the authors state, only associations can be drawn from their findings, not causal relationships – it might be that frailty causes a decline in swallow function]

Abstract

Similarly, it’s too bold a statement based on these results alone that “swallowing rehabilitation should be incorporated into frailty prevention programs..”

Round 2

Reviewer 2 Report

Thank you for the modifications. The interpretation of the results has improved. Only a couple of minor grammatical/language components to address in the amendments:

Line 286: "....would be underlie in older adults." I'm not sure what the authors mean here.

Line 332: "...Further intervention trials were needed to identify...." Should be "...are needed..."